Mitochondrial DNA hyperdiversity and its potential causes in the marine periwinkle Melarhaphe neritoides (Mollusca: Gastropoda)

Fourdrilis Séverine severine.fourdrilis@gmail.com sfourdrilis@naturalsciences.be 1
Mardulyn Patrick 2
Hardy Olivier J. 2
Jordaens Kurt 3
de Frias Martins António Manuel 4
Backeljau Thierry 1 5
1 Directorate Taxonomy and Phylogeny & JEMU, Royal Belgian Institute of Natural Sciences , Brussels , Belgium
2 Evolutionary Biology and Ecology, Université Libre de Bruxelles , Brussels , Belgium
3 Department of Biology, Invertebrate Section, Royal Museum for Central Africa , Tervuren , Belgium
4 CIBIO, Centro de Investigação em Biodiversidade e Recursos Genéticos, InBIO Laboratório Associado, Pólo dos Açores, Departamento de Biologia da Universidade dos Açores, University of the Azores , Ponta Delgada , Portugal
5 Evolutionary Ecology Group, University of Antwerp , Antwerp , Belgium
Collins Tim
Electronic publication date: 2016 Oct 5
Publication date: 2016
Volume: 4
Electronic Location ID: e2549
Received 2016 Jun 23; Accepted 2016 Sep 12
Copyright: ©2016 Fourdrilis et al.
Copyright year: 2016
Copyright holder: Fourdrilis et al.
License: This is an open access article distributed under the terms of the Creative Commons Attribution License, which permits unrestricted use, distribution, reproduction and adaptation in any medium and for any purpose provided that it is properly attributed. For attribution, the original author(s), title, publication source (PeerJ) and either DOI or URL of the article must be cited.
License URL: https://creativecommons.org/licenses/by/4.0/

Keywords: mtDNA hyperdiversity, Haplotype diversity, Nucleotide diversity, Planktonic dispersal, Effective population size, Selection, Mutation rate

Funding: Belgian federal Science Policy Office BELSPO Action 1 MO/36/027 Research Foundation –Flanders (FWO) W0.009.11N Royal Belgian Institute of Natural Sciences (RBINS) This research was funded by the Belgian federal Science Policy Office (BELSPO Action 1 MO/36/027). It was conducted in the context of the Research Foundation –Flanders (FWO) research community “Belgian Network for DNA barcoding” (W0.009.11N) and the Joint Experimental Molecular Unit (JEMU) at the Royal Belgian Institute of Natural Sciences (RBINS). The funders had no role in study design, data collection and analysis, decision to publish, or preparation of the manuscript.

==============================
We report the presence of mitochondrial DNA (mtDNA) hyperdiversity in the marine periwinkle Melarhaphe neritoides (Linnaeus, 1758), the first such case among marine gastropods. Our dataset consisted of concatenated 16S-COI-Cytb gene fragments. We used Bayesian analyses to investigate three putative causes underlying genetic variation, and estimated the mtDNA mutation rate, possible signatures of selection and the effective population size of the species in the Azores archipelago. The mtDNA hyperdiversity in M. neritoides is characterized by extremely high haplotype diversity (Hd = 0.999 ± 0.001), high nucleotide diversity (π = 0.013 ± 0.001), and neutral nucleotide diversity above the threshold of 5% (πsyn = 0.0677). Haplotype richness is very high even at spatial scales as small as 100m2. Yet, mtDNA hyperdiversity does not affect the ability of DNA barcoding to identify M. neritoides. The mtDNA hyperdiversity in M. neritoides is best explained by the remarkably high mutation rate at the COI locus (μ = 5.82 × 10−5 per site per year or μ = 1.99 × 10−4 mutations per nucleotide site per generation), whereas the effective population size of this planktonic-dispersing species is surprisingly small (Ne = 5, 256; CI = 1,312–3,7495) probably due to the putative influence of selection. Comparison with COI nucleotide diversity values in other organisms suggests that mtDNA hyperdiversity may be more frequently linked to high μ values and that mtDNA hyperdiversity may be more common across other phyla than currently appreciated.

Introduction

The term DNA hyperdiversity is usually applied to populations when neutral nucleotide diversity at selectively unconstrained synonymous sites is ≥5% (Cutter, Jovelin & Dey, 2013), that is when two 100 bp protein-coding DNA sequences (mitochondrial or nuclear) chosen randomly from a population sample differ on average at five or more synonymous and neutral nucleotide positions. Nucleotide diversity in a sequence alignment is calculated either from pairwise differences at all sites (π) or at segregating sites only (θ) (Nei, 1987; Nei & Miller, 1990; Watterson, 1975). Yet, π is often preferred because its estimation is less sensitive to sequencing errors and DNA sequence length than θ (Johnson & Slatkin, 2008). Nucleotide diversity is also calculated at synonymous sites (πsyn) to obtain an estimate of neutral polymorphism reflecting the balance between mutation pressure and genetic drift. This latter measure of πsyn is required to observe hyperdiversity. DNA hyperdiversity is usually associated with fast evolving prokaryotes and viruses and less frequently with eukaryotic organisms showing lower rates of evolution (Drake et al., 1998). Nevertheless, mitochondrial (mtDNA) or nuclear (nDNA) DNA data retrieved from literature references on 505 animal species, showed signatures of DNA hyperdiversity (πsyn ≥ 0.05) in 43% of the species studied, i.e., 42% among 394 Chordata, 55% among 66 Arthropoda, 33% among 24 Mollusca, 24% among 17 Echinodermata, and 100% among 3 Nematoda (Table S1). Although these percentages most probably reflect strong sampling bias, DNA hyperdiversity seems not uncommon in eukaryotes. Rates of mtDNA evolution are 10–30 times faster than nDNA and drive mitonuclear coevolution and speciation through strong selection pressure (Blier, Dufresne & Burton, 2001; Hill, 2016; Lane, 2009). Hyperdiverse intraspecific mtDNA variation provides a greater density of polymorphic sites for selection to act upon (Cutter, Jovelin & Dey, 2013), and possibly provokes higher speciation rate as observed in birds and reptiles (Eo & DeWoody, 2010). Studying mtDNA hyperdiversity is hence interesting to better understand how evolutionary processes such as mutational dynamics and selection that underlie mitonuclear coevolution contribute to speciation (Burton & Barreto, 2012).

mtDNA is a popular population genetic marker because of its variability and, as such, is widely used for evolutionary studies at the species level (Féral, 2002; Wan et al., 2004) and DNA barcoding (Hebert, Ratnasingham & DeWaard, 2003). The main determinants of animal mtDNA diversity are supposed to be mutation rate (μ) and selection, while in contrast to nDNA, effective population size (Ne) and ecology (life history traits) are expected to be less important (Bazin, Glémin & Galtier, 2006; Cutter, Baird & Charlesworth, 2006; Dey et al., 2013; Lanfear, Kokko & Eyre-Walker, 2014; Leffler et al., 2012; Nabholz, Glémin & Galtier, 2009; Nabholz et al., 2008; Small et al., 2007). The higher nDNA diversity observed in invertebrates vs. vertebrates, in marine vs. non-marine species, and in small vs. large organisms, is probably therefore not in line with patterns of mtDNA diversity (Bazin, Glémin & Galtier, 2006; Leffler et al., 2012). As the main determinants of animal mtDNA diversity are supposed to be μ and selection, mtDNA hyperdiversity is more likely to be also explained by high μ or selection on the mitochondrial genome, and we expect that the relationship between Ne and mtDNA diversity may be weakened. Still, at least in eutherian mammals and reptiles mtDNA diversity seems to correlate with Ne (Hague & Routman, 2016; Mulligan, Kitchen & Miyamoto, 2006), so that an eventual influence of Ne on mtDNA hyperdiversity cannot a priori be neglected. Hence, in summary, mtDNA diversity may be affected by amongst others: (1) mutations generating new alleles that increase mtDNA diversity, (2) diversifying and balancing selection that increase mtDNA diversity by favoring extreme or rare phenotypes (Maruyama & Nei, 1981; Mather, 1955; Rueffler et al., 2006), (3) other types of selection that decrease mtDNA diversity by eliminating disadvantageous alleles (Anisimova & Liberles, 2012), and (4) fluctuations in Ne since more mutations arise in populations with larger Ne (Kimura, 1983). Yet, far more empirical data are needed to better understand the relative contribution of various determinants of mtDNA hyperdiversity.

In the present work, we investigate three potential determinants of mtDNA hyperdiversity i.e., μ, selection and Ne, in the marine periwinkle Melarhaphe neritoides (Linnaeus, 1758) in the Azores archipelago. Melarhaphe neritoides is an intertidal gastropod that shows signatures of mtDNA hyperdiversity (see data in García et al., 2013). It is a small (shell up to 11 mm) temperate species (Lysaght, 1941), in which the sedentary adults produce pelagic egg capsules and long-lived planktonic larvae with high dispersal potential during 4–8 weeks until settlement (Cronin, Myers & O’riordan, 2000; Fretter & Manly, 1977; Lebour, 1935). Melarhaphe neritoides is widely distributed throughout Europe (Fretter & Graham, 1980), where it shows a remarkable macrogeographic population genetic homogeneity (inferred from allozyme data) (Johannesson, 1992), though locally in Spain it displays huge amounts of mtDNA COI diversity in terms of a large numbers of polymorphic sites (S = 16%), a very high haplotype diversity (Hd = 0.998) and a very high nucleotide diversity (π = 0.019) (García et al., 2013). We studied mtDNA diversity of M. neritoides within the archipelago of the Azores because this area provides a vast, though relatively isolated, setting to explore geographic mtDNA variation at different spatial scales.

First, we formally describe and evaluate mtDNA hyperdiversity in M. neritoides, by assessing diversity in three mtDNA gene fragments, viz. 16S ribosomal RNA (16S), cytochrome oxidase c subunit I (COI) and cytochrome b (Cytb) in substantial numbers of individuals and locations. Second, we survey the literature to compare M. neritoides mtDNA hyperdiversity with other littorinids, other planktonic-dispersing gastropods showing high genetic diversity, and other hyperdiverse molluscs in general. Finally, we explore the relationship between mtDNA diversity in M. neritoides and (1) μ, (2) selection, (3) Ne, (4) population genetic structuring, and (5) phylogeny.

Materials and Methods

Samples and DNA collection

A total of 610 specimens of M. neritoides were collected between 1992 and 2012 at six localities in the Azores archipelago, Portugal, viz. Varadouro, Faial island (FAI), Fajã Grande, Flores island (FLO), Mosteiros, São Miguel island (MOS), Lajes do Pico, Pico island (PIC), Maia, Santa Maria island (SMA), and Porto Formoso, São Miguel island (SMI) (Fig. 1). These 610 specimens contribute to our analyzed data sets as follows (Table S2): (1) dataset 1: 185 specimens from five islands sequenced for COI (614 bp), 16S (482 bp) and Cytb (675 bp) to investigate mtDNA diversity and demographic history; (2) dataset 2: 223 specimens from one island collected at a single spot of about 100 m2 at MOS and sequenced for COI (657 bp) and 213 among these sequenced for 16S (482 bp), to assess microscale mtDNA haplotype richness; (3) dataset 3: 169 specimens from four islands collected between 1992 and 1993, and 175 specimens collected in 2012 at the same four localities, sequenced for COI (578 bp) to generate a temporal series of samples over 20 years for estimating mtDNA μ; (4) dataset 4: 212 specimens from five islands sequenced for COI (605 bp), completed by one COI sequence of M. neritoides from the United Kingdom retrieved from GenBank (AJ488608) and 86 COI sequences of seven species from the three littorinid subfamilies Lacuninae, Laevilitorininae and Littorininae (Reid, Dyal & Williams, 2012) and one species of Pomatiidae available in GenBank, viz. Bembicium auratum (Lacuninae) (AJ488606), Cremnoconchus syhadrensis (Lacuninae) (AJ488605), Lacuna pallidula (Lacuninae) (AJ488604, KT996151), Laevilitorina caliginosa (Laevilitorininae) (AJ488607), Littorina littorea (Littorininae) (AJ622946, HM884235, HM884236, HM884248, KF643337, KF643416, KF643449, KF643454, KF643456, KF643464, KF643631, KF643658, KF643697, KF643729, KF643906, KF644042, KF644180, KF644262, KF644330), Peasiella isseli (Littorininae) (HE590849), Pomatias elegans (Pomatiidae) (JX911283, JQ964789, GQ424199, EU239237, EU239238, EU239239, EU239240, EU239241) and Tectarius striatus (Littorininae) (DQ022012 –DQ022064), to assess monophyly, possible phylogenetic structuring, and eventual cryptic taxonomic diversity in M. neritoides.

Figure 1 Sampling sites (cross-shaped symbols) of M. neritoides in the Azores archipelago, Portugal.

FAI, Varadouro, Faial island; FLO, Fajã Grande, Flores island; MOS, Mosteiros, São Miguel island; PIC, Lajes do Pico, Pico island; SMA, Maia, Santa Maria island; SMI, Porto Formoso, São Miguel island.

Collected specimens were preserved at −20 °C until DNA analysis. Individual genomic DNA was extracted from foot muscle following the standard protocol of either the NucleoSpin® Tissue kit (Macherey-Nagel GmbH & Co. KG, Düren, Germany) or the DNeasy 96 Blood & Tissue kit (Qiagen GmbH, Hilden, Germany). Remaining soft body parts and shells have been deposited in the collections of the Royal Belgian Institute of Natural Sciences, Brussels (RBINS) under the general inventory number IG 32962.

16S, COI and Cytb amplification and sequence alignment

PCR amplification was carried out in a 20-µL reaction volume using standard Taq DNA polymerase (Qiagen GmbH, Hilden, Germany) and universal primers LCO1490 (5′-GGTCAACAAATCATAAAGATATTGG-3′) and HCO2198 (5′-TAAACTTCAGGGT GACCAAAAAATCA-3′) for a 578-to-657 bp region of COI (Folmer et al., 1994), universal primers 16Sar (5′-CGCCTGTTTAACAAAAACAT-3′) and 16Sbr (5′-CCGGTCTGAA CTCAGATCACGT-3′) for a 482 bp region of 16S (Simon et al., 1994), and littorinid-specific primers 14825 (5′-CCTTCCCGCACCTTCAAATC-3′) and 15554 (5′-GCAAATAAAAAG TATCACTCTGG-3′) for a 675 bp region of Cytb (Reid, Rumbak & Thomas, 1996). The PCR conditions for COI consisted of an initial denaturation at 95 °C for 5 min, 40 cycles of denaturation at 95 °C for 45 s, annealing at 45 °C for 45 s, elongation at 72 °C for 1 min 30 s, and a final elongation at 72 °C for 10 min. The PCR conditions for Cytb were the same except for the annealing step at 48 °C. The PCR conditions for 16S were also the same except for the annealing step at 52 °C, 35 cycles instead of 40, and final elongation for 5 min. PCR products were purified using Exonuclease I and FastAP Thermosensitive Alkaline Phosphatase (Thermo Scientific, Erembodegem-Aalst, Belgium). Sequencing reactions were performed directly on purified PCR products using the BigDye® Terminator v1.1 Cycle Sequencing kit (Life Technologies, Gent, Belgium) and run on an Applied Biosystems 3130xl Genetic Analyser automated capillary sequencer, or outsourced to Macrogen (Rockville, MD, USA). Sample files were assembled, edited and reviewed using ABI Prism® SeqScape® 2.5.0 (Applied Biosystems). The accuracy and reproducibility of the PCR results were validated by triplicating COI and 16S amplifications on a subset of 20 individuals, using standard Taq DNA polymerase for two replicates and HotStar HiFidelity DNA Polymerase (Qiagen GmbH, Hilden, Germany) for one replicate. Sequence alignments were made with ClustalW (Thompson, Higgins & Gibson, 1994) using default parameters in BioEdit 7.0.9.0 (Hall, 1999). All sequences were deposited in GenBank (KT996151 –KT997344). The morphology-based identification of M. neritoides was validated through DNA barcoding by querying the 185 COI fragments from dataset 1 in the Barcode of Life Data systems (BOLD) (Ratnasingham & Hebert, 2007).

mtDNA diversity

The three gene fragments were concatenated for the 185 specimens of dataset 1, using Geneious 5.3.4 (http://www.geneious.com, Kearse et al., 2012). DNA diversity metrics (Tables 1 and 2) were calculated with DnaSP 5.10.1 (Librado & Rozas, 2009). Despite the fact that 32–42 specimens were sampled per site, there were no shared haplotypes between sampling sites (Hs = 0), i.e., all haplotypes were private (Table 1). Consequently, we examined mtDNA haplotype richness at a microscale, i.e., within a sampling site. Dataset 2, composed of identical fragment lengths across individuals, was used to compute individual-based rarefaction curves for the COI and 16S fragments using EstimateS 8.2.0 (Colwell, 2006) in order to assess the relationship between the number of haplotypes observed (Hobs) and sample size, and compute the Chao1 and Chao2 richness estimators (Chao, 1984; Chao, 1987). Given that COI and Cytb showed similar diversity levels (Table 1), only COI was used for rarefaction analysis. A logarithmic trendline, best fitting the data, was applied to each rarefaction curve to extrapolate Hobs for larger sample sizes.

Table 1 mtDNA diversity metrics of Melarhaphe neritoides.

Statistics describing the number of individuals (N), number of haplotypes (H), number of private haplotypes (Hp), number of shared haplotypes among sampling sites (Hs), number of shared haplotypes within sampling site (Hw), DNA fragment length in base pairs (L), number of segregating sites (S) and its corresponding percentage of the fragment length into brackets, haplotype diversity (Hd) ± standard deviation, Jukes-Cantor corrected nucleotide diversity (π) ± standard deviation, Jukes-Cantor corrected nucleotide diversity at synonymous sites (πsyn) and Jukes-Cantor corrected nucleotide diversity at non-synonymous sites (πnon-syn).

	N	H	Hp	Hs	Hw	L	S	Hd	π	πsyn	πnon-syn	
16S-COI-Cytb	185	184	184	0	1	1,771	420 (24%)	0.999 ± 0.001	0.013 ± 0.001	0.0677	0.0004	
16S	185	77	63	12	2	482	71 (15%)	0.814 ± 0.030	0.004 ± 0.001	–	–	
COI	185	156	142	13	1	614	169 (28%)	0.996 ± 0.002	0.018 ± 0.001	0.0736	0.0001	
Cytb	185	166	153	9	4	675	180 (27%)	0.998 ± 0.001	0.016 ± 0.001	0.0637	0.0006	

Table 2 Overview of mtDNA diversity in other Littorinidae, various highly diverse planktonic-dispersers and hyperdiverse mollusc species.

Taxa are listed by decreasing value of haplotype diversity.

Species		Larval development	Sampling area	N	Locus	L	Hd	π	Reference	
Other Littorinidae	
Temperate species	
Littorina saxatilis	Mo	d	North Atlantic	453	ND1-tRNApro-ND6-Cytb	1,154	0.940	0.005	Doellman et al. (2011)	
			North Atlantic	778	Cytb	607	0.905	0.009	Panova et al. (2011)	
Tectarius striatus	Mo	p (unknown)	Macaronesia	109	COI-Cytb	993	0.934	0.006	Van den Broeck et al. (2008)	
Littorina keenae	Mo	p (unknown)	North Pacific	584	ND6-Cytb	762	0.815	0.003	Lee & Boulding (2007)	
Littorina littorea	Mo	p (28–42 days)	North Atlantic	488	COI	424	0.810	0.004	Calculated from data in Wares et al. (2002), Williams, Reid & Littlewood (2003), Williams & Reid (2004), Giribet et al. (2006), Blakeslee, Byers & Lesser (2008), Layton, Martel & Hebert (2014)	
Littorina plena	Mo	p (64 days)	NE Pacific	135	Cytb	414	0.775	0.006	Lee & Boulding (2009)	
Littorina obtusata	Mo	d	North Atlantic	46	COI	582	0.762	0.006	Calculated from data in Wares & Cunningham (2001)	
			NW Atlantic	31	COI	574	0.127	0.001	Calculated from data in Layton, Martel & Hebert (2014)	
Bembicium vittatum	Mo	d	Indian Ocean	40	12S	324	0.730	–	Kennington, Hevroy & Johnson (2012)	
Austrolittorina unifasciata	Mo	p (4 weeks)	Australia	102	COI	658	0.541	0.002	Calculated from data in Colgan et al. (2003), Williams, Reid & Littlewood (2003), Waters, Mcculloch & Eason (2007)	
Littorina scutulata	Mo	p (37–70 days)	NE Pacific	265	Cytb	414	0.389	0.003	Lee & Boulding (2009)	
Littorina subrotundata	Mo	d	NE Pacific	229	Cytb	414	0.297	0.001	Lee & Boulding (2009)	
Austrolittorina antipodum	Mo	p (4 weeks)	New Zealand	40	COI	658	0.146	0.001	Calculated from data in Williams, Reid & Littlewood (2003), Waters, Mcculloch & Eason (2007)	
Littorina sitkana	Mo	d	NE Pacific	146	Cytb	414	0.093	0.001	Lee & Boulding (2009)	
Tropical species	
Echinolittorina reticulata	Mo	p (3–4 weeks)	Indo-Pacific	37	COI	1,251	1.000	0.009	Reid et al. (2006)	
Echinolittorina vidua	Mo	p (3–4 weeks)	Indo-Pacific	92	COI	1,217	0.996	0.041	Reid et al. (2006)	
Echinolittorina trochoides C	Mo	p (3–4 weeks)	Indo-Pacific	14	COI	1,251	0.989	0.006	Reid et al. (2006)	
Littoraria coccinea glabrata	Mo	p (unknown)	Indian Ocean	45	COI	451	0.954	0.006	Silva et al. (2013)	
Echinolittorina trochoides A	Mo	p (3–4 weeks)	Indo-Pacific	46	COI	1,251	0.943	0.009	Reid et al. (2006)	
Echinolittorina trochoides B	Mo	p (3–4 weeks)	Indo-Pacific	18	COI	1,251	0.935	0.004	Reid et al. (2006)	
Bembicium nanum	Mo	p (weeks)	Australia	54	COI	806	0.920	0.006	Ayre, Minchinton & Perrin (2009)	
Echinolittorina trochoides E	Mo	p (3–4 weeks)	Indo-Pacific	21	COI	1,251	0.900	0.003	Reid et al. (2006)	
Echinolittorina trochoides D	Mo	p (3–4 weeks)	Indo-Pacific	20	COI	1,251	0.884	0.003	Reid et al. (2006)	
Cenchritis muricatus	Mo	p (4 weeks)	Caribbean	77	COI	282	0.850	0.008	Díaz-Ferguson et al. (2012)	
Echinolittorina ziczac	Mo	p (3–4 weeks)	Caribbean Sea	31	COI	431	0.750	0.004	Díaz-Ferguson et al. (2012)	
Echinolittorina lineolata	Mo	p (3–4 weeks)	South Atlantic	496	COI	441	0.704	0.003	Calculated from Genbank data KJ857561 –KJ858054 and Williams & Reid (2004)	
			South Atlantic	442	Cytb	203	0.284	0.002	Calculated from Genbank data KM210838 –KM211279	
Littoraria scabra	Mo	p (unknown)	Indo- Pacific	50	COI	527	0.690	0.003	Silva et al. (2013)	
Littoraria irrorata	Mo	p (4 weeks)	NE Atlantic	238	COI	682	0.546	0.004	Calculated from data in Díaz-Ferguson et al. (2010), Robinson et al. (2010), Reid, Dyal & Williams (2010)	
Other highly diverse planktonic-dispersing marine invertebrates	
Glaucus atlanticus	Mo	p (lifelong)	Worldwide	112	COI	658	0.996	0.014	calculated from data in Churchill, Alejandrino & Valdés (2013), Churchill, Valdés & Ó Foighil (2014), Wecker et al. (2015)	
Pygospio elegans	An	p (4–5 weeks)	North Sea	23	COI	600	0.996	0.014	Kesäniemi, Geuverink & Knott (2012)	
Argopecten irradians concentricus	Mo	p (5–19 days)	NW Atlantic	219	mtDNA	1,025	0.982	0.008	Marko & Barr (2007)	
Brachidontes pharaonis	Mo	p (weeks)	Mediterranean-Red Sea	34	COI	618	0.973	0.039	Terranova et al. (2007)	
Ruditapes philippinarum	Mo	p (2–3 weeks)	NW Pacific	170	COI	644	0.960	0.010	Mao et al. (2011)	
Cellana sandwicensis	Mo	p (4 days)	Hawaii	109	COI	612	0.960	0.006	Bird et al. (2007)	
Holothuria nobilis	Ec	p (13–26 days)	Indo-Pacific	360	COI	559	0.942	0.008	Uthicke & Benzie (2003)	
Tridacna maxima	Mo	p (9 days)	Indo-Pacific	211	COI	484	0.940	0.023	Nuryanto & Kochzius (2009))	
Tridacna crocea	Mo	p (1 week)	Indo-Malaysia	300	COI	456	0.930	0.015	Kochzius & Nuryanto (2008))	
Pachygrapsus crassipes	Ar	p (95 days)	NE Pacific	346	COI	710	0.923	0.009	Cassone & Boulding (2006)	
Tripneustes gratilla	Ec	p (18 days)	Indo-Pacific	83	COI	573	0.902	0.004	calculated from data in Lessios, Kane & Robertson (2003)	
Holothuria polii	Ec	p (13–26 days)	Mediterranean Sea	158	COI	484	0.873	0.005	Vergara-Chen et al. (2010)	
Nacella magellanica	Mo	p (unknown)	SW Atlantic	171	COI	573–650	0.868	0.004	Aranzamendi, Bastida & Gardenal (2011)	
Bursa fijiensis	Mo	p (8 weeks)	SW Pacific	59	COI	566	0.848	0.003	Castelin et al. (2012)	
Acropora cervicornis	Cn	p (4 days)	Caribbean	160	mtCR	941	0.847	0.006	Vollmer & Palumbi (2007)	
Other hyperdiverse mollusc species	
Pliocardia kuroshimana	Mo	p	NW Pacific	3	mtDNA	513	1.000	0.256*	James, Piganeau & Eyre-Walker (2016)	
Bulinus forskalii	Mo	–	–	12	mtDNA	339	1.000	0.167*	James, Piganeau & Eyre-Walker (2016)	
Pyrgulopsis intermedia	Mo	d	–	15	mtDNA	528	0.924	0.148*	James, Piganeau & Eyre-Walker (2016)	
Euhadra brandtii	Mo	n/a	–	14	mtDNA	558	0.989	0.098*	James, Piganeau & Eyre-Walker (2016)	
Biomphalaria glabrata	Mo	d	–	7	mtDNA	579	0.714	0.092*	James, Piganeau & Eyre-Walker (2016)	
Achatinella mustelina	Mo	n/a	–	69	mtDNA	675	0.992	0.078*	James, Piganeau & Eyre-Walker (2016)	
Quincuncina infucata	Mo	d	–	5	mtDNA	453	1.000	0.067*	James, Piganeau & Eyre-Walker (2016)	
Pyrgulopsis thompsoni	Mo	d	–	7	mtDNA	657	0.952	0.066*	James, Piganeau & Eyre-Walker (2016)	
Notes.

An Annelida

Ar Arthropoda

Cn Cnidaria;

Ec Echinodermata

Mo Mollusca

d direct larval development

p planktonic larval development (pelagic larval duration given in parenthesis)

n/a not applicable

N number of individuals

L locus length in base pairs

Hd haplotype diversity

π nucleotide diversity

– missing data.

* π calculated at synonymous sites only (πsyn).

Population genetic structure

The monophyly of M. neritoides was assessed, and p-distances were compared within and among clades, in order to detect possible cryptic taxa and/or phylogenetic structuring that might contribute to the overall mtDNA hyperdiversity. First, two species trees were produced from dataset 4 using Bayesian inference (BI) and Maximum Likelihood (ML). Seven Littorinidae species were added to the ingroup. The outgroup Pomatias elegans belongs to a different family (Pomatiidae), but the same superfamily (Littorinoidea) as M. neritoides. Two independent runs of BI were performed using MrBayes 3.2.2 (Ronquist et al., 2012) hosted on the CIPRES Science Gateway (Miller, Pfeiffer & Schwartz, 2010), under a GTR + G nucleotide substitution model selected according to jModelTest 2.1.4 (Darriba et al., 2012), for 4.106 generations with a sample frequency of 100 and a 30% burn-in. Convergence between the two runs onto the stationary distribution was assessed by examining whether the potential scale-reduction factors was close to 1 in the pstat file, standard deviation of split frequencies fell below 0.01 in the log file, and trace plots showed no trend by examining the p files in TRACER 1.6 (Rambaut et al., 2014). The final consensus tree was computed from the combination of both runs. ML analysis based on the GTR+G model was conducted in MEGA 6.06 (Tamura et al., 2013), with bootstrap consensus trees inferred from 1,000 replicates. Second, three methods of species delimitation were used: (1) the Automatic Barcode Gap Discovery (ABGD, available at http://wwwabi.snv.jussieu.fr/public/abgd/abgdweb.html) method (Puillandre et al., 2012), (2) the Bayesian implementation of the Poisson tree Processes (bPTP, available at http://species.h-its.org/ptp/) model (Zhang et al., 2013), and (3) the General Mixed Yule Coalescent (GMYC, available at http://species.h-its.org/gmyc/) model (Fujisawa & Barraclough, 2013; Pons et al., 2006). Finally, sequence divergence within and between clades was assessed by calculating mean within-group p-distances for the four species comprising more than one sequence (Littorina littorea, N = 19; M. neritoides, N = 213; Pomatias elegans, N = 8; Tectarius striatus, N = 53), and mean between groups p-distances for all species pairs (Table S2), using MEGA. Additionally, COI sequence divergence within M. neritoides was assessed by generating an intraspecific p-distances distribution from the 185 COI sequences included in dataset 1, using MEGA.

Dataset 1 was subjected to the program ALTER (http://sing.ei.uvigo.es/ALTER/, Glez-Peña et al., 2010) to convert the Fasta-formatted sequence alignment to a sequential Nexus-formatted file, which then could be analyzed by NETWORK 4.6.1.2 (www.fluxus-engineering.com, Bandelt, Forster & Röhl, 1999) to reconstruct a median-joining haplotype network. Population genetic structure in M. neritoides was qualitatively investigated with the haplotype network which provides information about phylogeographic structure and gene flow among populations, and quantified by GST (Pons & Petit, 1995), NST based on a distance matrix of pairwise differences (Pons & Petit, 1996) and φST (Excoffier, Smouse & Quattro, 1992) using dataset 1 in SPAGEDI 1.4 (Hardy & Vekemans, 2002) for GST and NST and ARLEQUIN 3.5.1.3 (Excoffier & Lischer, 2010) for φST.

mtDNA mutation rate

mtDNA evolves fast enough to provide sufficient variation for the estimation of μ over a two-decades period (Drummond et al., 2003), i.e., the time span of our temporal sampling and corresponding to 5–6 generations of M. neritoides. Dataset 3 comprises different sampling points in time, allowing sequences to be treated as heterochronous data for estimating the number of mutations occurring in the time interval between samples as described in Seo et al. (2002) and Drummond et al. (2002). In this way, the mutation rate per nucleotide site per year can be inferred using a Bayesian MCMC method as implemented in BEAST 2.1.3 (Bouckaert et al., 2014) hosted on the CIPRES Science Gateway. The Bayesian MCMC analysis was performed under a HKY substitution model (the closest model to GTR since GTR is not available in BEAST) with empirical base frequencies and a fixed substitution rate of 1.0 and a tree prior set to “coalescent exponential population” (chosen after model comparison with the “coalescent constant population” and the “coalescent Bayesian skyline” priors), a strict clock model assuming a constant substitution rate over time and a prior set to lognormal with M = − 5 and S = 1.25. The analysis was run in triplicate for 500 million generations with a sample frequency of 50,000 and 10% burn-in. Convergence of MCMC chains was assessed by visual examination of the log trace of each posterior distribution showing caterpillar shape in TRACER, and making sure that the Effective Sample Size (ESS) value of each statistic was >200 (Ho & Shapiro, 2011). The three runs were combined using LOGCOMBINER 2.1.3 (part of the BEAST package) and the final ESSs were at least 1,100. The estimate of μ was provided under the “Estimates” tab in TRACER as the mean of the “clockRate” parameter.

Demography, selection and effective population size

Departure from mutation-drift equilibrium indicative of demographic change or selective sweep was assessed in dataset 1 using Tajima’s D (Tajima, 1989) and Fu’s Fs (Fu, 1997) tests implemented in ARLEQUIN and 10,000 coalescent-based simulations were run to calculate p-values. Since Tajima’s D and Fu’s Fs statistics are sensitive to both demographic change and selection, we also applied Fay & Wu’s H statistic (Fay & Wu, 2000) to dataset 1 for the single 16S, COI, Cytb fragments and the concatenated 16S-COI-Cytb data using DnaSP to attempt to discriminate between the effects of population size change and selection (Zeng et al., 2006). Tectarius striatus was the most closely related species to M. neritoides (Reid, Dyal & Williams, 2012) for which the three same gene fragments of 16S, COI and Cytb were available on Genbank (U46825, AJ488644, U46826), and was therefore used as outgroup for the Fay & Wu test. Confidence intervals were calculated based on 10,000 coalescent-based simulations.

ARLEQUIN was used to construct a distribution of pairwise nucleotide differences between haplotypes (sequence mismatch distribution) and to compare this distribution with the expectations of a sudden expansion model (Harpending, 1994; Li, 1977; Rogers, 1995). Although the analysis complied with the assumption of panmixis, it did not do so with respect to neutrality (see ‘Results’), thus limiting the reliability of the results. Three demographic parameters were inferred using a generalized nonlinear least-squares method to determine whether M. neritoides has undergone sudden population growth: the rate of population growth τ = 2μt (t being the time since the expansion), the initial population size before the growth (θ0), and the final population size after growth (θ1). The goodness of fit between the observed and expected mismatch distributions was tested by parametric bootstrapping of the sum of squared deviations (Ssd) (Schneider & Excoffier, 1999), and by the Harpending Raggedness index (r) (Harpending et al., 1998).

A time-calibrated Bayesian skyline plot (BSP) was built for dataset 1 using BEAST, to detect past population dynamics through time and to estimate Ne of M. neritoides in the Azores. The coalescent priors used in the skyline plot model assume a random sample of orthologous, non-recombining and neutrally evolving sequences from a panmictic population. The skyline plot model has been shown to be robust to violation of these assumptions and to correctly reconstruct demographic history with mtDNA (Drummond et al., 2005). However, recent studies show that violation of these assumptions may still affect the estimated population size variation, and that the BSP is prone to confound the effect of population structure with declines in population size in panmictic populations, or fails to detect population expansion (Grant, 2015; Heller, Chikhi & Siegismund, 2013). Hence, population structure and selection were assessed beforehand using Tajima’s D, Fu’s Fs and Fay & Wu’s H statistics. The BSP analyses were performed under a HKY substitution model with empirical base frequencies, a fixed substitution rate equal to 1.0, and a piecewise-constant Bayesian skyline model with 10 groups. The prior on the clockRate parameter was set to a log-normal with M = − 5 and S = 1.25. Analyses were run in triplicate for 200 million generations with a sample frequency of 20,000 and 10% burn-in. After combination of the three runs, the final ESSs were at least 1,000. Ne was extracted from the BSP, by dividing the median value of the Ne∗τ product in the most recent year 1996 (Ne∗τ≈17,977) by the generation time τ = 3 years and five months (Hughes & Roberts, 1981).

Results

mtDNA diversity of Melarhaphe neritoides

Dataset 1, representing the overall population of the Azores archipelago (N = 185 from five localities in the archipelago), contains 184 different and private haplotypes (H = 184; Hp = 184 and hence Hs = 0) (Table 1), except for one haplotype that was found in two individuals from Pico island (Hw = 1). Hence, the frequency of this latter, i.e., the most common, haplotype was 0.0108, while all other haplotypes had a frequency of 0.00541. This remarkable mtDNA diversity is further reflected by a haplotype diversity (based on the concatenated 16S-COI-Cytb data) close to its maximum value 1 (Hd = 0.999 ± 0.001), indicating a probability of less than 0.001% that two individuals from the same locality share the same haplotype in the overall population of the archipelago. One fourth of the 1771 nucleotide positions are polymorphic (S = 23.7%) with 167 sites (9.4%) showing one variant, 225 sites (12.7%) two variants, 24 sites (1.4%) three variants, and four sites (0.2%) four variants. Moreover, there is on average 1.3% nucleotide differences per site between two randomly chosen DNA sequences in the overall population (π = 0.013 ± 0.001). More precisely, among the protein-coding COI and Cytb regions (1,289 bp), we observed 323 (18%) sites with synonymous and 964 (54%) sites with non-synonymous substitutions, yielding on average πsyn = 0.0677(6.77%) and πnon−syn = 0.0004(0.04%) at synonymous and non-synonymous sites respectively. Repeating COI and 16S PCR amplifications on 20 specimens yielded identical sequence results, confirming that PCR did not generate artificial variation. For 185 query sequences of M. neritoides submitted to BOLD, that stores 51 barcodes of M. neritoides (data retrieved from BOLD on 13 October 2015), the identification engine returns 100% correct identifications under one and the same Barcode Index Number (BIN = BOLD:AAG4377).

Figure 2 Individual-based rarefaction curves (solid lines) and 95% confidence intervals (dashed lines) based on COI, 16S and concatenated 16S-COI data, based on M. neritoides specimens sampled in Mosteiros (MOS), São Miguel island.

Hobs is the haplotype richness observed in the actual sample (n) from MOS. The logarithmic trendlines (dotted lines) show a prediction of the haplotype richness expected for larger sampling size at the MOS sampling site.

Figure 3 Mismatch distribution analysis showing the unimodal distribution of the observed number of differences between pairs of haplotypes of M. neritoides.

Ssd, sum of squared differences and p-value in parenthesis; r, Harpending’s Raggedness index and p-value in parenthesis; τ, time in generations since the last demographic expansion; θ0, initial population size; θ1, final population size.

The individual-based rarefaction curves of 16S, COI and 16S-COI (dataset 2) do not reach a plateau, but their steep slopes decrease according to 16S-COI > COI > 16S (Fig. 2). Hobs values are close to the maximal sampling size n for the COI (Hobs = 180, n = 223) and the 16S-COI (Hobs = 174, n = 197) fragments indicating that a large fraction of the haplotype diversity remains to be discovered, whereas it is further from n for the 16S fragment (Hobs = 71, n = 213). The logarithmic trendlines representative of the population growth in the species show inflexion around large sampling sizes (n > 500), indicating that additional sampling is likely to yield new haplotypes. Indeed, the Chao1 (chao1 mean = 1596.22, CI = [878.94–3043.34], n = 197) and Chao2 (chao2 mean = 1486.20, CI = [842.91–2748.15], n = 197) estimators for the concatenated 16S-COI gene fragment suggest that the predicted total haplotype richness of M. neritoides would be reached by sampling 1,500 individuals per sampling site.

Demography, selection and mutation rate

Both Tajima’s D and Fu’s Fs tests show a significant departure of M. neritoides from constant population size or neutrality (D = − 2.030, p < 0.01 and Fs = − 23.706, p < 0.01), suggesting demographic expansion and/or a potential action of selection. Fay & Wu’s H, which is sensitive to positive selection and not to population growth or background selection, shows significant signal of selection for 16S (H = − 30.42, CI = [−3.06–1.13]), COI (H = − 85.16, CI = [−17.05–6.08]), Cytb (H = − 110.38, CI = [−13.33–5.18]) and the concatenated 16S-COI-Cytb fragment (H = − 225.96, CI = [−30.36–11.22]). The unimodal curve of the sequence mismatch distribution (Fig. 3) suggests that population expansion cannot be rejected as θ0 < θ1 (τ = 25.543, θ0 = 4.366, θ1 = 123.516). The non-significant values of the sum of squared deviations (Ssd = 0.00118, p = 0.600) and Harpending’s Raggedness index (r = 0.0005, p = 0.998) show that the sudden expansion model provides a good fit to the data. The time-calibrated BSP shows an increase of Ne through time, indicating that M. neritoides has been expanding in the Azores archipelago or has undergone selection (Fig. 4). For the year 1996, the BSP gives Ne∗τ≈17,977, corresponding to Ne ranging from 1,312 to 37,495 with an average Ne ≈ 5,256 individuals.

Figure 4 Historical demographic trends of the median estimate of the maternal effective population size over time (bold line) constructed using a Bayesian skyline plot approach based on concatenated 16S-COI-Cyt b haplotypes of M. neritoides sampled in 1992, 1993 and 1996.

The y-axis is the product of effective population size (Ne) and generation time (τ) in a log scale, while the x-axis is a linear scale of time in years. The 95% highest probability density (HPD) intervals are shaded in grey and represent both phylogenetic and coalescent uncertainty.

With data sampled in 1992, 1993 and 2012 (i.e., an interval of 20 years), we estimated a mutation rate of μ = 5.82 × 10−5 per nucleotide site per year at COI. Considering a generation time of τ = 41 months (i.e., 3.42 years), the mutation rate was estimated to be μ = 1.99 × 10−4 mutations per nucleotide site per generation.

Population genetic structure

All phylogenetic trees provided maximal support for the monophyly of M. neritoides (trees not shown). Additionally, the three species delimitation methods, ABGD, bPTP and GMYC, lumped M. neritoides as one Molecular Operational Taxonomic Unit (trees not shown). The mean intraspecific p-distance within M. neritoides was d = 0.018 ± 0.002, i.e., one order of magnitude greater than the mean intraspecific p-distances of the three other species, viz. Littorina littorea (d = 0.004 ± 0.001), Pomatias elegans (d = 0.009 ± 0.002) and Tectarius striatus (d = 0.006 ± 0.001), but still far below interspecific p-distances ranging from 0.166 to 0.271 for the 36 possible species pairs of Littorinoidea, from 0.166 to 0.246 for the 21 species pairs of Littorinidae, or from 0.187 to 0.225 for the six species pairs of Littorininae (Table S2). The Gaussian distribution of intraspecific COI p-distances in M. neritoides (Fig. 5) indicates that the five populations sampled on five different islands of the Azores archipelago form a homogeneous haplotype mixture without any evidence of a DNA barcode gap.

The bush-like pattern of the mtDNA haplotype network (Fig. 6) shows the overwhelming number of unique, private haplotypes represented by single individuals (i.e., singletons), the lack of shared haplotypes between sites, and several homoplastic character states (cycles). The apparent lack of association between genetic variation and geographic location (as revealed by the distribution of colours across the network of Fig. 6) suggests the absence of phylogeographic structure in Azorean M. neritoides.

Figure 5 Distribution of COI pairwise p-distances in M. neritoides.

Figure 6 Median-joining network of mtDNA in M. neritoides.

Branch lengths are proportional to the numbers of mutational steps separating haplotypes. The size of circles is proportional to the number of individuals per haplotype and the sole haplotype shared by two individuals is marked by an arrow. Haplotype origins: Flores island—green; Faial island—blue; Pico island—yellow; São Miguel island—red; Santa Maria island—purple.

The low and non-significant indices of population genetic differentiation (GST = 0.0003, p = 0.1676; NST = 0.0021, p = 0.5346; φST = 0.0026, p = 0.2220) make that the hypothesis of panmixis (and hence no population structuring) cannot be rejected.

Discussion

How diverse is the mtDNA of Melarhaphe neritoides?

Azorean M. neritoides harbours a remarkable amount of intraspecific mtDNA diversity, characterized by very high haplotype diversity and nucleotide diversity with respect to the concatenated 16S-COI-Cytb gene fragments, at the single Cytb gene fragment and at the single COI gene fragment. Moreover, it shows a value of neutral mtDNA nucleotide diversity πsyn ≥ the threshold of 5% for the concatenated 16S-COI-Cytb fragments, and is therefore qualified as hyperpolymorphic. The πsyn values for COI and Cytb separately are also ≥ 0.05 and support mtDNA hyperdiversity in M. neritoides (Table 1). mtDNA hyperdiversity is also observed in a Spanish population. The COI data retrieved from García et al. (2013) yielded πsyn = 0.0762 (7.62%) and πnon−syn = 0.0002 (0.02%) in a local Spanish population of 49 individuals. These values are very similar to those of COI in the Azorean populations (Table 1). Therefore, mtDNA hyperdiversity is not a local characteristic of M. neritoides along the Iberian Atlantic coast, but is shared more broadly in the Azorean populations, and presumably, throughout the species’ distribution range. The high π values in M. neritoides reflect natural variation, not PCR errors, as validated by the identical triplicates of mtDNA sequences and 100% correct species identification using barcoding. DNA barcoding is based on the premise that COI sequence divergence is higher among species than within species (Hebert, Ratnasingham & DeWaard, 2003), and might be hampered by high mtDNA variation, specifically COI hyperdiversity and high intraspecific sequence divergence in COI. Yet, in spite of the highly variable COI marker in M. neritoides (π = 0.018 ± 0.001) and elevated intraspecific p-distance (d = 0.001–0.041), the ability of DNA barcoding to identify M. neritoides is not affected by this mtDNA hyperdiversity.

The mtDNA of M. neritoides is more diverse than (1) mtDNA of most temperate littorinids and many tropical littorinids, (2) mtDNA of many planktonic-dispersing marine invertebrates, and (3) mtDNA of other hyperdiverse Mollusca (Table 2). More specifically, in comparison with 26 other littorinid species, M. neritoides has the highest COI haplotype diversity among temperate species (i.e., Austrolittorina spp., Bembicium vittatum, Littorina spp., Tectarius striatus) and the same degree as two tropical species Echinolittorina reticulata and Echinolittorina vidua. Melarhaphe neritoides also has the highest COI nucleotide diversity among temperate species, and shows a higher COI nucleotide diversity than tropical species (i.e., Bembicium nanum, Cenchritis muricatus, Echinolittorina spp., Littoraria spp.) except for Echinolittorina vidua whose nucleotide diversity (π = 0.041) is about twice that of M. neritoides (π = 0.018). In comparison to 15 other non-littorinid marine invertebrates with similar planktonic larval dispersal and high mtDNA variability, M. neritoides has the highest COI haplotype diversity. Yet, M. neritoides shows the same degree of COI haplotype diversity as the pelagic nudibranch Glaucus atlanticus (Hd = 0.996) and the annelid Pygospio elegans (Hd = 0.996). Regarding COI nucleotide diversity, M. neritoides has the highest value among annelids, arthropods, cnidarians, echinoderms, other gastropods, and some bivalves (but not all). Two bivalves, viz. Brachidontes pharaonis and Tridacna maxima, show very high COI nucleotide diversities that probably reflect ongoing speciation in the three lineages of the Brachidontes spp. complex (Terranova et al., 2007) and in the four lineages in Tridacna maxima (Nuryanto & Kochzius, 2009). The literature data in Table 2 suggest that there is no obvious correlation between π and Hd. However, more data are needed to corroborate this observation.

We estimated the neutral component of the COI nucleotide hyperdiversity in M. neritoides, i.e., πsyn = 0.074, on which the diagnosis of hyperdiversity is based. In comparison to eight other mollusc species with hyperdiverse mtDNA (Table 2), M. neritoides is situated in the lower part of the neutral nucleotide diversity range (πsyn = [0.066–0.256]).

mtDNA divergence and population structuring in Melarhaphe neritoides

We investigated whether population genetic structure through time and space, and cryptic taxa, could contribute to the mtDNA hyperdiversity in M. neritoides. The monophyly of M. neritoides and the Gaussian distribution of its intraspecific p-distances, suggest that M. neritoides does not conceal cryptic taxa in the Azores. Conversely, the overwhelming number of private haplotypes (Fig. 6) at first glance suggest that populations are strongly differentiated because of the apparent lack of shared haplotypes. Yet, the bush-like mtDNA haplotype network (Fig. 6) is suggestive of complete population mixing (Nielsen & Slatkin, 2013). Indeed, recurrent long-term gene flow homogenising the gene pool of M. neritoides over the 600 km between the Azorean islands implies an absence of population genetic structure (differentiation), as is reflected in the GST, NST and φST values that are not significantly different from zero. This is congruent with the low level of differentiation and high potential for long range gene flow between Swedish and Cretan populations of M. neritoides (Johannesson, 1992). Currently, no other data on population genetic differentiation and gene flow in M. neritoides are available. The possibility of long-distance gene flow may suggest that the mtDNA diversity of M. neritoides in the Azores is the result of larval influx from European populations. Yet, while short-lived Pleistocene westward-flowing sea surface currents allowed the colonization of the Azores from Eastern Atlantic areas (Ávila et al., 2009), the eastward-flowing Azores Current nowadays (Barton, 2001) suggests that larval transport predominantly occurs from the Azores towards the North East Atlantic coasts and the Mediterranean Sea, and that the Azores rather may act as a source of new, dispersing, haplotypes than as a sink receiving new haplotypes. Hence, all current evidence suggests that mtDNA hyperdiversity in M. neritoides is not due to (1) population structuring, (2) admixture of divergent local populations, (3) lumping of cryptic taxa, or (4) influx of new haplotypes from distant European populations.

mtDNA mutation rate in Melarhaphe neritoides

We investigated whether mtDNA mutation rate explains mtDNA hyperdiversity. The mutation rate is the rate at which new mutations arise in each generation of a species and accumulate per DNA sequence, and differs from the substitution rate that accounts for the fraction of new mutations that do not persist in the face of evolutionary forces (Barrick & Lenski, 2013). Accordingly, neutral synonymous mutations reflect the mutation rate (Barrick & Lenski, 2013). Mutation rates in most nuclear eukaryotic genomes are generally extremely low because elaborate molecular mechanisms correct errors in DNA replication and repair DNA damage, whereas viral and animal mitochondrial genomes have no, or far less efficient, repair mechanisms and thus have much higher mutation rates (Ballard & Whitlock, 2004; Drake et al., 1998). Overall, synonymous mutations become fixed at a rate that appears to be uniform across various taxa (Kondrashov, 2008), and mtDNA mutation rates lie in a narrow range of 10−8–10−7 mutations per nucleotide site per generation across e.g., arthropods, echinoderms, chordates, molluscs and nematodes (Table 3). Surprisingly, our estimate of the mtDNA COI mutation rate in M. neritoides (μ = 1.99 × 10−4 per site per generation) is 1,000 to 10,000-fold higher than commonly estimated for the mtDNA mutation rates in metazoans from these phyla. Therefore, if our inference is correct, it seems likely that this high mtDNA mutation rate substantially contributed to generating the mtDNA hyperdiversity in M. neritoides. Our mutation rate estimate was obtained from mtDNA sequence data of M. neritoides itself, not from closely related species, and is therefore expected to be more accurate and species-specific. Bayesian MCMC estimates of substitution rates based on heterochronous mtDNA samples may be susceptible to an upward bias when populations have a complex demographic history (e.g., bottleneck) or pronounced population structure. Hence, such biased estimates may reflect other processes like migration, selection and genetic drift rather than mutation (Navascués & Emerson, 2009). However, this study did not provide evidence of population structure in M. neritoides, reducing therefore the risk of bias in the estimate of μ. Bayesian MCMC inferences based on heterochronous mtDNA samples over short timescales may also overestimate generational mutation rates by an order of magnitude in comparison to phylogenetically derived mutation rates, because they may account for short-lived, slightly deleterious mutations at non-synonymous sites (Ho et al., 2005; Penny, 2005; Subramanian & Lambert, 2011). Since μ in M. neritoides was estimated over a short period of 20 years, it may be subject to such a bias. However, while this bias could have generated an order of magnitude overestimation of μ, it cannot entirely account for the extreme value inferred, which is 103 to 104 fold higher than usually estimated for other organisms (Subramanian & Lambert, 2011).

Table 3 mtDNA mutation rates per site per generation in various metazoans ranked according to decreasing μ.

Species		μ	locus	Reference	
Melarhaphe neritoides	Mo	1.99 × 10−4	COI	this study	
Homo sapiens sapiens	Ch	6.00 × 10−7	mt genome	Kivisild (2015)	
Caenorhabditis elegans	Ne	1.60 × 10−7	mt genome	Denver et al. (2000)	
Mytilus edulis	Mo	9.51 × 10−8	COI	Wares & Cunningham (2001)	
Drosophila melanogaster	Ar	6.20 × 10−8	mt genome	Haag-Liautard et al. (2008)	
Asteria rubens	Ec	4.84 × 10−8	COI	Wares & Cunningham (2001)	
Nucella lapillus	Mo	4.43 × 10−8	COI	Wares & Cunningham (2001)	
Euraphia spp.	Ar	3.80 × 10−8	COI	Wares & Cunningham (2001)	
Idotea balthica	Ar	3.60 × 10−8	COI	Wares & Cunningham (2001)	
Semibalanus balanoides	Ar	2.76 × 10−8	COI	Wares & Cunningham (2001)	
Littorina obtusata	Mo	2.49 × 10−8	COI	Wares & Cunningham (2001)	
Sesarma spp.	Ar	2.10 × 10−8	COI	Wares & Cunningham (2001)	
Alpheus spp.	Ar	1.90 × 10−8	COI	Wares & Cunningham (2001)	
Prochilodus spp.	Ch	0.27 × 10−8	COI	Turner et al. (2004)	
Notes.

Ar Arthropoda

Ch Chordata

Ec Echinodermata

Mo Mollusca

Ne Nematoda

Invertebrates with shorter generation times have higher mtDNA mutation rates, as their mitochondrial genomes are copied more frequently (Thomas et al., 2010). In comparison to the generation times of invertebrates analyzed by Thomas et al. (2010), ranging from 8 days in the hydrozoan Hydra magnipapillata to 1,825 days in the coral Montastraea annularis and the seastar Pisaster ochraceus, the generation time of M. neritoides (τ ≈ 1,250 days) is not particularly short and therefore its mtDNA mutation rate would be expected to be at the lower side. Yet, M. neritoides has a high mtDNA mutation rate (μ = 5.82 × 10−5 per site per year) that does not fall within the range of mutation rates of invertebrates with longer generation times than M. neritoides, i.e., from μ = 3 × 10−10 per site per year in Montastraea annularis (Fukami & Knowlton, 2005) to μ = 2.81 × 10−6 per gene per year in Pisaster ochraceus (Popovic et al., 2014).

High mtDNA mutation rates may be more frequently linked to hyperdiversity than previously thought in the widely used COI marker. Indeed, neutral nucleotide diversities of ≥0.05 have been reported in 222 other species among Arthropoda, Chordata, Echinodermata, Mollusca and Nematoda (Table S1), suggesting the possibility of underlying high mtDNA mutation rates.

Demography and selection in Melarhaphe neritoides

We investigated whether selection, mtDNA demographic history and Ne explain mtDNA hyperdiversity. Equilibrium between variation gained by mutations and variation lost by genetic drift should be reached if the effective population size has been stable over time and in absence of population structure or selection (Kimura, 1983). According to the negative Tajima’s D, Fu’s Fs and Fay & Wu’s H, the unimodal sequence mismatch distribution and the BSP trend, the phylogeny of Azorean M. neritoides has been shaped either by demographic expansion or selection, or a combination of both.

The effective mtDNA population size of M. neritoides estimated in this paper is Ne ≈ 5, 256 (CI = 1,312–37,495) for the concatenated 16S-COI-Cytb gene fragments. This is relatively small in comparison to mtDNA Ne of other littorinids with planktonic larval stages and high dispersal potential like Littorina plena (Ne = 16,0526–33,728,571) and Littorina scutulata (Ne = 90,790–3,814,286) (Table 4), except for the mtDNA Ne in the planktonic dispersing Littorina keenae (Ne = 135) (Lee & Boulding, 2007). Yet, this latter value refers to one sampling site only, whereas another sampling site of Littorina keenae showed a much larger mtDNA Ne (Ne = 31,797). Surprisingly, and somewhat counterintuitively, the mtDNA of M. neritoides is also smaller than that of periwinkles without planktonic larval stages, such as Littorina sitkana (Ne = 105,263–1,400,000) and Littorina subrotundata (Ne = 25,000 –1,942,857) (Lee & Boulding, 2009). However, past putative selection in M. neritoides likely confounds the BSP inference by reducing the overall mtDNA diversity and thus the mtDNA Ne estimate. As such, mtDNA variation in M. neritoides is still remarkably high, despite this signal of a reduction of its diversity by the putative influence of selection. This strengthens the hypothesis that the mtDNA hyperdiversity in M. neritoides is best explained by a high μ of mtDNA.

Table 4 mtDNA effective population sizes (Ne) for various taxa.

The 95% confidence interval is given in parenthesis when available.

Taxon		Ne	Locus	Reference	
Littorina keenae	Mo	135 (42–2,490)	ND6-Cytb	Lee & Boulding (2007)	
Melarhaphe neritoides	Mo	5,256 (1,312–37,495)	COI-16S-Cytb	this study	
Homo & Pan	Ch	5,900–10,000	mt genome	Piganeau & Eyre-Walker (2009)	
Felidae & Canidae	Ch	130,000–430,000	mt genome	Piganeau & Eyre-Walker (2009)	
Pachygrapsus crassipes	Ar	167,000–1,020,000	COI	Cassone & Boulding (2006)	
Cardinalis cardinalis	Ch	193,000 (4,000–701,000)	ND2-Cytb	Smith & Klicka (2013)	
Murinae	Ch	230,000–730,000	mt genome	Piganeau & Eyre-Walker (2009)	
Littorina sitkana	Mo	105,263–1,400,000	Cytb	Lee & Boulding (2009)	
Littorina subrotundata	Mo	25,000–1,942,857	Cytb	Lee & Boulding (2009)	
Littorina scutulata	Mo	90,790–3,814,286	Cytb	Lee & Boulding (2009)	
Littorina plena	Mo	160,526–33,728,571	Cytb	Lee & Boulding (2009)	
Notes.

Ar Arthropoda

Ch Chordata

Mo Mollusca

mtDNA Ne and mtDNA hyperdiversity may be positively correlated such as in the lined shore crab Pachygrapsus crassipes (Ne = 167,000–1,020,000; COI Hd = 0.923; π = 0.009) (Cassone & Boulding, 2006). Yet, this relationship has been questioned (Bazin, Glémin & Galtier, 2006; Piganeau & Eyre-Walker, 2009), because Bazin, Glémin & Galtier (2006) showed that mtDNA diversity is not linked to mtDNA Ne, but rather to μ and selection. Conversely, Nabholz, Glémin & Galtier (2009) and Nabholz et al. (2008) found no link between selection and mtDNA Ne, but confirmed that mtDNA diversity is strongly linked to μ. Our present work shows a link between mtDNA hyperdiversity and high mtDNA μ, and the putative influence of selection on Ne estimation making mtDNA Ne a poor indicator of mtDNA hyperdiversity.

Conclusions

The mtDNA hyperdiversity of M. neritoides is characterized by a high haplotype diversity (Hd = 0.999 ± 0.001), a high nucleotide diversity (π = 0.013 ± 0.001) and a high neutral nucleotide diversity (πneu = 0.0678) for the concatenated 16S-COI-Cytb gene fragments. The mutation rate at the COI locus is μ = 1.99 × 10−4 mutations per nucleotide site per generation, which is a very high value. Demographic analyses revealed that M. neritoides in the Azores underwent a population expansion, but the effective population size Ne was surprisingly small for a planktonic-developing species (Ne = 5, 256; CI = 1,312–37,495) probably due to the putative influence of selection on M. neritoides mtDNA. As a result, Ne is not linked to mtDNA hyperdiversity and is a poor indicator of this latter. Mitochondrial DNA hyperdiversity is best explained by a high mtDNA μ in M. neritoides. Mitochondrial DNA hyperdiversity may be more common across eukaryotes than currently known.

Supplemental Information

Table S1 List of 215 animal species with hyperdiverse DNA (πsyn > 0.05)

Click here for additional data file.

Table S2 Specimen samples and data sets used in this study

N, number of individuals; N1, N2, N3, N4, number of individuals used in dataset 1, dataset 2, dataset 3 and dataset 4 respectively.

Click here for additional data file.

Table S3 Estimates of average evolutionary divergence over COI sequence pairs within and between groups

n, number of sequences used; d1, number of base differences per site (p-distance) from averaging over all sequence pairs within each group ± standard error; n/c, cases in which it was not possible to estimate evolutionary distances; d2, number of base differences per site (p-distance) from averaging over all sequence pairs between groups (under diagonal) and standard error estimates (above diagonal).

Click here for additional data file.

We are grateful to Vanya Prévot and Marine Monjardez (both RBINS) for their help with DNA sequencing. We are indebted to David Reid (Natural History Museum, London) for providing Lacuna pallidula. We thank Thierry Hoareau (University of Pretoria, South Africa) for helpful comments about BSP analysis, Peter Marko (Clemson University, USA) for mtDNA sequences of Argopecten irradians concentricus, and Jonathan Waters (University of Otago, New Zealand) for COI sequences of Austrolittorina sp.

Additional Information and Declarations

Competing Interests

Author Contributions

DNA Deposition

Data Availability

The authors declare there are no competing interests.

Séverine Fourdrilis conceived and designed the experiments, performed the experiments, analyzed the data, wrote the paper, prepared figures and/or tables.

Patrick Mardulyn and Olivier J. Hardy reviewed drafts of the paper.

Kurt Jordaens contributed analyses discussion.

António Manuel de Frias Martins contributed reagents/materials/analysis tools.

Thierry Backeljau conceived and designed the experiments, contributed reagents/materials/analysis tools, reviewed drafts of the paper.

The following information was supplied regarding the deposition of DNA sequences:

New sequences are available from Genbank with accession numbers from KT996151 to KT997344.

The following information was supplied regarding data availability:

Dataset 1: https://dx.doi.org/10.6084/m9.figshare.3437333.v4;

Dataset 2 (16S): https://dx.doi.org/10.6084/m9.figshare.3437390.v2;

Dataset 2 (COI): https://dx.doi.org/10.6084/m9.figshare.3437474.v2;

Dataset 2 (16S_COI): https://dx.doi.org/10.6084/m9.figshare.3437489.v2;

Dataset 3: https://dx.doi.org/10.6084/m9.figshare.3437498.v2;

Dataset 4: https://dx.doi.org/10.6084/m9.figshare.3437531.v2.

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
