# Peer review of "Mitochondrial DNA hyperdiversity and its potential causes in the marine periwinkle Melarhaphe neritoides (Mollusca: Gastropoda)"

_PeerJ, doi:10.7717/peerj.2549_

## Round 0.1 · original submission · Minor Revisions

In this well written manuscript the authors present the results of a
thoughtful and careful study of mitochondrial DNA (mtDNA) hyperdiversity in populations of the marine gastropod Melaraphe neritoides sampled in the Azores. Their main conclusions are that in this case the likely cause of mtDNA hyperdiversity is the exceptionally high rate of nucleotide evolution at the COI locus, and that mtDNA hyperdiversity across phyla many be more common than is currently acknowledged.

Both external reviewers viewed the manuscript very favorably and recommended that the manuscript be accepted after minor revisions, and I agree with their assessment. Their comments and suggestions are clearly spelled out and should be addressed in the manuscript, and in your cover letter for the revision, so don't need to be rehashed here in detail. I do think reviewer one has a good point that the connectivity and relationship of the Azores populations to continental populations, especially those in the study by García et al. 2013, although I disagree that figure 1 should be dropped, as it shows the geography of sampled populations. Reviewer two questions whether it is possible to infer that there is no genetic structure given that virtually all haplotypes are unique. While this is true if we treat haplotypes non-herarchically, it seems to me that your haplotype network (figure 6) can only be explained by haplotypes movement among islands, which seems unsurprising, given the planktonic larvae of this species.

Line 258: I think something is missing here. Maybe: it did not do so

Reviewer 1 ·

Basic reporting

This is a well planned, well executed, and well written study that addresses the phenomenon of hyperdiversity and its potential causes in the marine periwinkle Melarhaphe neritoides. In the Introduction to the paper, the authors do an excellent job of defining what is meant by hyperdiversity and explaining that, although not widely recognized in eukaryotes, this may be a relatively common phenomenon. The authors clearly outline the three chief objectives of the study: to investigate the putative causes of this hyperdiversity with respect to mutation rate, selection, or changes in effective population size. Although their results do not provide definitive conclusions, the authors address their objectives in a logical fashion and attempt to assess evidence supporting each putative cause. Based on comparisons to other taxa, they suggest that hyperdiversity in this species is best explained by its extraordinarily high mutation rate (inferred from their data). In the process, they do an excellent job of documenting hyperdiversity within Azorean populations of this species, as well as placing their results in the broader context of other molluscan species and planktonic dispersers.

With respect to the geographic context of this study, the authors state that they chose the Azores to base their study because this area provides a vast, though relatively isolated, setting to explore geographic mtDNA variation at different spatial scales. Although this is certainly a good rationale, at several times while reading this paper I was interested in knowing more about the genetic connectivity (e.g. extent and direction of gene flow) of Azorean populations to other European populations. This was not discussed, however. Also I was expecting the authors to compare their results to Spanish populations which have also been reported to displays high levels of mtDNA COI diversity (García et al. 2013). Other than a brief reference to this in the Introduction, these populations were never referred to again. Broadening the scope of analysis and discussion to geographic regions beyond the Azorean populations would be useful and informative.

Experimental design

The authors rely on four mtDNA datasets collected on Melarhaphe neritoides over a 20 year period. The number of samples and geographic distribution of sampling associated with each dataset is illustrated effectively in Supplementary Table S2. The inclusion of datasets sampled across years is particularly useful for assessing the mtDNA mutation rate and simulating the demographic history of populations of this species. The authors effectively describe how each dataset is used to address the varied analyses undertaken. Appropriate controls to test the reliability of PCR and sequencing were performed. Methods and analyses appear appropriate, with discussion of relevant assumptions and instances where those assumptions are likely to be robust (or not) to violations.

Validity of the findings

The authors have done well to attempt to elucidate the role of mutation, population size and selection, and don’t over extend their results beyond what their data appear to show. Clearly there is evidence to indicate the hyperdiversity of mtDNA may be linked to an elevated mutation rate, particularly in comparison to other taxa. The authors also suggest that the mtDNA hyperdiversity of this species may have been influenced by a recent demographic expansion, possibly in combination with the effect of strong selection (lines 483-485). While the influence of elevated mutation rates on mtDNA hyperdiversity is reasonably intuitive, the mechanics of how demographic changes or effects of strong selection (reducing mtDNA diversity) have ultimately resulted in a taxon with high mtDNA diversity are less clear. Hence it would be useful for the authors to explain these processes more fully. Likewise, with respect to elevated mutation rates, it would be interesting to know if there are any other hallmarks of a high mutation rate that could be elucidated in the mtdna genome. For instance, could mtDNA gene rearrangements be associated with elevated mutation rates? Has the mtDNA genome of this species been completely sequenced to determine if there any other unusual aspects of its mtDNA?

Additional comments

This is a really interesting study that is presented in a very well written and well researched paper.

As discussed above, I think that it would be very useful to provide some commentary concerning what is known about gene flow between Azorean populations and other European populations. The authors refer to the relatively long larval duration of this species in the plankton, but don’t provide details (if known) about larval transport, current patterns, and possible immigration of new haplotypes into the Azores from elsewhere. The authors clearly state that the focus of this study is the Azores, but are the Azores genetically isolated from other European populations or is there a regular influx of new larval recruits from distant European populations (e.g. based on genetic studies, simulations, drogue studies)? Could this in any way help to explain the high mtDNA diversity in the Azores? Given the fact that hyperdiversity has also been reported in Spanish populations of this species, explanatory causes of this phenomenon have to be viewed in a broader context than just the Azores.

The authors include a provocative statement in their Introduction that “hyperdiverse intraspecificDNA variation provides a greater density of polymorphic sites and more statistical power to detect shifts in polymorphism and determine the molecular evolutionary processes underlying genetic differentiation (Cutter et al. 2013).” I think that it might be worth elaborating on this a little in the Discussion. Clearly demonstrating that hyperdiversity is more common that currently recognized is very important, as is understanding its causative factors. Demonstrating a practical application for using “model hyperdiverse taxa” such as Melarhaphe neritoides to better understand fundamental evolutionary processes is also very exciting and may help to broaden the interest of this paper to evolutionary biologists.

Other comments:
There is inconsistent formatting of papers in the Literature Cited section. Check the use of capitals in some articles and not others.

Tables.

The tables provided were very well laid out and extremely useful and informative.
I was curious about one aspect of Table 2 where the authors chose to make comparisons with other hyperdiverse “planktonic-dispersers”. By making this specific comparison to other planktonic dispersers, the authors appear to be making a link hyperdiversity and planktonic-dispersal. If so, this does not come through in the text of the article and may be worth commenting on.

Figures.
The figure formatting is a little basic in places and could use some finessing to improve the value of the figures to the reader such as by improving the labeling of values along axes.
Figure 1 is probably not needed. In Figure 2, more detailed labeling of values along each axis would be useful. Figure 3: the title of this figure needs to be reworked since it does not help the reader interpret the figure. Figure 5: the binning of p-distances along the x-axis seems odd here. I realize that this might be to emphasize the lack of a barcode gap. Likewise it is difficult to match the occurrence category to the axis value.

Supplementary tables: All are useful.

Reviewer 2 ·

Basic reporting

no comments

Experimental design

Very sound: well sampled at a range of temporal and spatial scales, well described. Careful confirmation of sequencing accuracy is appreciated in the context of this dataset.

Validity of the findings

Findings are valid with respect to the hyperdiversity of this taxon and the authors do an excellent and thorough job of placing this within the context of what is known about hyperdiversity in molluscs and animals in general. My only concerns are with respect to their conclusions about how their findings may be influenced by population genetic structure, which they conclude is absent. As noted in my general comments to authors, I don't think their data support this conclusion, so this needs to be reconsidered, but I don't think that one aspect alters the general validity of the study.

Additional comments

Introduction:

Lines 60-61: As essential background, another line or two about the Table S1 study would be appreciated and helpful: 43% of species, based on what initial sample -- certainly not of all known species in those 5 phyla! All species with a certain level of representation in GenBank, or...? Also, it seems worth noting that nearly all the examples in Table S1 are chordates; the proportion of invertebrates is very small, and of marine invertebrates, extremely small I think. This imbalance is worth noting - and perhaps a hypothesis could be formulated: is this pattern likely to be phylogenetic, or habitat-driven (terrestrial vs aquatic/marine)? It would help to set up the novelty of the present study to highlight how few examples of hyperdiverse marine invertebrates there are, even in highly speciose groups like molluscs well represented in bar-coding studies.

Lines 68, 75: It would help the reader to explain why selection on mtDNA would produce hyperdiversity; I am used to thinking about selection as reducing diversity, and would expect it act in opposition to mutation rate where genetic diversity is concerned. A line of explanation would be helpful to put this in context here.

Methods: Given that the introduction discusses how synonymous sites are typically examined to assess mtDNA hyperdiversity, it was a bit confusing to see the authors analyze 16S which is not protein-coding. Some explanation would be helpful to clarify why this is an appropriate marker given the typical focus on synonymous sites, or to make it clear why this marker does not represent a departure from past studies.

Including MP and NJ trees is overkill for testing monophyly in this context and I would remove those analyses to streamline the paper; it takes focus off the main issues of this paper and those analyses are weak compared to the model-based ML and BI analyses included.

Some better justification for choice of outgroups (including genera selected, and why) would help firm up the monophyly analysis and screen for cryptic species; were there no more congeners available to use as ingroups to see if the presumptive study taxon was truly monophyletic? Also, if this is your hypothesis, then the formal way to test for monophyly is to perform an AU test in RaxML comparing the likelihood of an unconstrained topology versus one in which the presumptive focal taxon is forced to be monophyletic; I'm not sure that's strictly needed here, but if that's the hypothesis the authors really want to test, then nodal support with a random selection of 6-7 outgroups is not a test of monophyly for part of the ingroup.

Similarly, if you want a more comprehensive screen for the possible presence of cryptic species, I would recommend doing so explicitly, using some species delimitation method - ABGD, GYMC, BPP, etc. Do any of these coalescent-based methods split your nominal taxon into more than one species, or do they all see it as one hyperdiverse group that nevertheless coalesces into a single species? This could well be worth including, given the focus of this paper; these aren't difficult analyses to perform on a dataset like this. I am sure they will affirm the conclusions of the authors.

Discussion:
Lines 416-424: It is not correct to conclude that there is no population structure because Gst values are low. In a situation like this, where all individuals are genetically unique, it is mathematically guaranteed that any Gst or Fst-like statistic will be near zero, but at the same time, with no shared haplotypes between sites, neither is there any evidence for gene flow between subpopulations. Please review and reference the extensive discussion of this issue beginning with Jost (2008) and his Dest test statistic, which was created to deal with precisely this situation, Hedrick (2005)'s modified Gst statistic, and others [see Heller and Siegismund (2009) Molecular Ecology for a good overview of this debate]. A considerable literature now exists addressing how best to estimate population structure in situations where genetic hyperdiversity means most individuals carry unique haplotypes. I do not agree with the authors' assessment that there is no population structure in the Azores; I think with their data, this simply cannot be ruled out. All populations are fully differentiated at these mtDNA loci, so at a minimum, one cannot draw conclusions about differentiation using these markers. This is also important for the inferences made elsewhere in the discussion, i.e. lines 451-452 regarding mutation rate.

Most BSPs I have seen go back in time tens to hundreds of thousands of years; I have not typically seen plots that only extend back a few decades. Some methodological discussion of how or why this analysis was restricted to the period during which samples were collected might help the reader - was the analysis extended to consider the effects of a historical (Pleistocene?) bottleneck and expansion, for instance? The mismatch distribution looks like an expanding population, not an equilibrium population..

---

## Round 0.2 · accepted · Accept

The authors have done a nice job of responding to the points made by the external reviewers, and I believe this manuscript is ready for publication.